# Infective Endocarditis—Predictors of In-Hospital Mortality, 17 Years, Single-Center Experience in Bulgaria

**DOI:** 10.3390/microorganisms12091919

**Published:** 2024-09-21

**Authors:** Bistra Dobreva-Yatseva, Fedya Nikolov, Ralitsa Raycheva, Mariya Tokmakova

**Affiliations:** 1Section of Cardiology Cardiology Clinic, First Department of Internal Medicine, Faculty of Medicine, Medical University—Plovdiv, UMBAL “St. Georgi” EAD, 4000 Plovdiv, Bulgaria; fedya.nikolov@mu-plovdiv.bg (F.N.); maryia.tokmakova@mu-plovdiv.bg (M.T.); 2Department of Social Medicine and Public Health, Faculty of Public Health, Medical University—Plovdiv, 4000 Plovdiv, Bulgaria; ralitsa.raycheva@mu-plovdiv.bg

**Keywords:** infective endocarditis, in-hospital mortality, acute heart failure, septic shock, early surgery

## Abstract

Despite enormous developments in medicine, infective endocarditis (IE) remains an ongoing issue for physicians due to increased morbidity and persistently high mortality. Our goal was to assess clinical outcomes in patients with IE and identify determinants of in-hospital mortality. Material and methods: The analysis was retrospective, single-centered, and comprised 270 patients diagnosed with IE from 2005 to 2021 (median age 65 (51–74), male 177 (65.6%). Native IE (NVIE) was observed in 180 (66.7%), prosthetic IE (PVIE) in 88 (33.6%), and cardiac device-related IE (CDRIE) in 2 (0.7%), with non-survivors having much higher rates. Healthcare-associated IE (HAIE) was 72 (26.7%), Staphylococci were the most prevalent pathogen, and the proportion of Gram-negative bacteria (GNB) non-HACEK was significantly greater in non-survivors than survivors (11 (15%) vs. 9 (4.5%), *p* = 0.004). Overall, 54 (20%) patients underwent early surgery, with a significant difference between dead and alive patients (3 (4.5%) vs. 51 (25.1%, *p* = 0.000). The overall in-hospital mortality rate was 24.8% (67). Logistic regression was conducted on the total sample (*n* = 270) for the period 2005–2021, as well as the sub-periods 2005–2015 (*n* = 119) and 2016–2021 (*n* = 151), to identify any differences in the trend of IE. For the overall group, the presence of septic shock (OR-83.1; 95% CI (17.0–405.2), *p* = 0.000) and acute heart failure (OR—24.6; 95% CI (9.2–65.0), *p* = 0.000) increased the risk of mortality. Early surgery (OR-0.03, 95% CI (0.01–0.16), *p* = 0.000) and a low Charlson comorbidity index (OR-0.85, 95% CI (0.74–0.98, *p* = 0.026) also lower this risk. Between 2005 and 2015, the presence of septic shock (OR 76.5, 95% CI 7.11–823.4, *p* = 0.000), acute heart failure (OR-11.5, 95% CI 2.9–46.3, *p* = 0.001), and chronic heart failure (OR-1.3, 95% CI 1.1–1.8, *p* = 0.022) enhanced the likelihood of a fatal outcome. Low Charlson index comorbidity (CCI) lowered the risk (OR-0.7, 95% CI 0.5–0.95, *p* = 0.026). For the period 2016–2021, the variable with the major influence for the model is the failure to perform early surgery in indicated patients (OR-240, 95% CI 23.2—2483, *p* = 0.000) followed by a complication of acute heart failure (OR-72.2, 95% CI 7.5–693.6. *p* = 0.000), septic shock (OR-17.4, 95% CI 2.0–150.8, *p* = 0.010), previous stroke (OR-9.2, 95% CI 1.4–59.4, *p* = 0.020) and low ejection fraction (OR-1.1, 95% CI 1.0–1.2, *p* = 0.004). Conclusions: Knowing the predictors of mortality would change the therapeutic approach to be more aggressive, improving the short- and long-term prognosis of IE patients.

## 1. Introduction

Although IE was described 350 years ago [1], it continues to provide a significant challenge to doctors for a variety of reasons. IE is a developing illness, and despite the availability of advanced imaging and microbiological tools, identification is frequently challenging and delayed. The global incidence of IE has increased to 13/100,000 [2]. In recent decades, improvements in medical and surgical treatment have had no effect on death or major complication rates [2,3]. Prevention is critical for reducing morbidity in IE. Maintaining proper oral hygiene and dental health, as well as following current recommendations for dental care and other procedures and manipulations, are critical for preventing IE.

The mortality rate for IE remains high, ranging from 15 to 30% [2,3,4], and is linked to the disease’s shifting characteristics. Patients are older than sixty years and have numerous comorbidities. A high Charlson Comorbidity Index (>3), diabetes, renal failure, prosthetic valve IE (PVIE), and hemodialysis all predict a bad prognosis [4,5,6,7,8]. The prevalence of PVIE has increased in recent decades [2,9,10]. Novel types of IE, such as indwelling device-related IE and post-TAVI (transcatheter aortic valve implantation) IE, are becoming more common as medical science and technology develop. They are difficult to treat and have a poor prognosis because they occur in older individuals with severe comorbidities and frequently require surgery [11,12,13,14,15]. 

Healthcare-associated IE (HAIE) accounts for almost one-third of all cases and has been driven by medical advancements, improved treatment, and a higher life expectancy of patients. It is often associated with a poor outcome [16,17]. The microbiological spectrum of IE has shifted, with staphylococci and enterococci predominating, posing challenges for treatment [10]. 

Complications, such as acute or worsening heart failure, septic shock, acute neurological events, acute renal failure related to valve dysfunction, embolism, and/or persistent infection, are the most important predictors of mortality [18]. These are also the main indications for early surgical intervention.

Early risk assessment for complications and death is crucial. The timely identification of high-risk patients would shift the therapeutic approach to a more aggressive one, particularly when deciding on surgical therapy. Early surgery is associated with reduced in-hospital mortality and improved long-term prognosis for IE patients [9,10]. Knowing the predictors of death and complications can help us make better treatment options. 

The characteristics of IE vary according to the country’s geographical and socioeconomic status. Randomized prospective studies are difficult to undertake because the condition is relatively rare. For certain regions and nations, information is mostly collected via retrospective and single-center examinations.

There has been a shortage of information on the predictors of in-hospital mortality in Bulgaria over the past few decades. We looked at the determinants of in-hospital death in patients with IE over the course of 17 years.

## 2. Material and Method

This is a retrospective, single-center study, including 270 patients diagnosed with IE according to the modified Duke criteria, treated at the University Hospital “St Georgi” in the city of Plovdiv, Bulgaria for the period January 2005–December 2021. Patients were separated into two sub-periods: before (2005–2015, *n* = 119) and after (2016–2021, *n* = 151) the most recent current IE guidelines (2015) to find differences in the disease progression. The incidence has been growing during the last six years. We observed an increase in cases of prosthetic IE, healthcare-associated IE, increased age, and improved diagnosis. These findings are consistent with current IE data provided by the Global Burden of Disease 2019. 

The hospital’s capacity is 1500 beds, and the Cardiology Clinic is a reference center for treating IE for a large part of southern Bulgaria. The medical records of treated patients with codes I33, I38, and I39 for the described period were used. Variables studied included demographics, risk group, presence of predisposing heart disease, comorbidities, Charlson Comorbidity Index (CCI) [19], entry gate, predictors of transient bacteremia, clinical, echocardiographic findings, causative organisms, complications, and clinical outcome.

### 2.1. Definition and Classification of IE

The diagnosis was defined as definite or possible IE according to the modified Duke criteria [20]. Surgical treatment of IE was defined as early if surgery was performed during antibiotic treatment. Valvular involvement in IE was determined on the basis of echocardiographic findings, other imaging studies, cardiac surgery, or, in some cases, clinical presentation. IE was classified by mode of acquisition as community-associated IE (CAIE), healthcare-associated IE (HAIE), and intravenous drug-associated IE (IDUIE). These categories are mutually exclusive. IE was defined as HAIE according to the following criteria: occurrence of IE > 48 h after hospital admission or within 6 months after hospital discharge for ≥2 days; IE developed within 6 months after a significant invasive procedure performed during hospitalization or in an outpatient setting; extensive outpatient healthcare contacts, defined as receiving wound care or intravenous treatment within 1 month before the onset of IE; or stay in a clinic-home for similar care [12,13,14,15]. IE occurring on a prosthetic valve within 12 months of surgery is defined as prosthetic valve early endocarditis (PVIE) and is classified as HAIE. Patients with a recent (within 1 month) or longer history of intravenous drug use were classified as IDUIE. Patients with no medical history and no history of injecting drug use were classified as CAIE. IE following dental treatment was considered to be CAIE if there was no other healthcare contact. The presence of septic emboli and an extracardiac focus of infection was defined as a focus of infection detected by imaging or on the basis of typical clinical presentation. Complications were diagnosed according to established diagnostic criteria and recommendations.

### 2.2. Statistical Methods

Quantitative data are presented as arithmetic mean ± standard deviation (mean ± SD) or median and interquartile range (25–75 percentiles) according to the type of distribution of the variables (Kolmogorov–Smirnov test). Categorical variables were summarized using absolute (*n*) and relative (%) magnitudes. A Mann–Whitney test for independent samples was used to compare quantitative variables between two groups. A Z-test was used to compare the relative shares of categorical variables between the studied groups. Logistic regression was performed to determine the simultaneous influence of significant independent variables identified in the univariate analysis to predict belonging to one of two mutually exclusive categories (alive/dead) of the dependent variable output. To determine whether there were differences in the distribution of survival between patients who underwent early surgery and those who did not, a log-rank test was performed. A *p*-value < 0.05 (two-tailed test) was considered statistically significant for all tests. A statistical analysis was performed using SPSS, version 26.0 (IBM Corp., Binghamton, NY, USA).

## 3. Results

Of the 270 patients, 205 (75.9%) had definite IE, with 133 (65%) having two major criteria and 72 (35%) having one major plus three minor criteria. There were 65 (24.1%) cases of possible IE, 62 (95%) with one major and one minor criterion, and three with three minor criteria. The overall mortality rate was 24.8% (67). Table 1 shows the baseline characteristics of the patients. The median age was 65 (51–74) years, with non-survivors being significantly older than survivors (67 (53–75) vs. 62 (44–73) years, *p* = 0.003). In the whole sample, there were 177 men (65.6%), with no difference between survivors and non-survivors. The majority of patients (180/66.7%) had native valve IE, whereas 88 (33.6%) had prosthetic valve IE. We found CDRIE only in the non-survivor group (2 (3%), *p* = 0.013) in patients with pacemaker, VVI mode (single chamber). There was no significant difference between deceased and surviving patients in terms of risk group, predisposing cardiac conditions, port of entry, or type of IE based on mode of acquisition. The most common congenital heart condition is bicuspid aortic valves and mitral valve prolapsus (7.1%), with two patients (0.7%) having repaired Tetralogy of Fallo. According to current guidelines (1998, 2007, 2015), they were all not eligible for antibiotic prophylaxis.

Patients had a wide range of comorbidities, the most common of which were arterial hypertension 171 (63.3%), chronic heart failure 124 (45.9%), previous cardiac surgery 95 (35.2%), chronic renal failure 70 (25.9%), coronary artery disease 64 (23.7%), diabetes 51 (18.9%), atrial fibrillation 49 (18.1%), and more. We discovered a significant difference with more cases in the deceased than in those who survived CCI (4 (3–6) vs. 3 (1–5), *p* < 0.0001), chronic renal disease (26 (38.8%) vs. 44 (24.7%), *p* = 0.006), atrial fibrillation (18 (26.9%) vs. 31 (15.3%), *p* = 0.033), and prior stroke (15 (22.4%) vs. 25 (12.3%), *p* = 0.044). The most prevalent symptoms were fever 263 (97.4%), anemia 248 (92.5%), and a heart murmur 178 (66.2%). The number of cases with splenomegaly 49 (8.1%) and rash 14 (5.5%) decreased dramatically. The deceased group had significantly fewer febrile cases (63 (94%) versus 200 (98.5%), *p* = 0.045). In terms of complications, we found that there were significantly more cases of acute heart failure (57 (85.1%) vs. 71 (35%), *p* < 0.0001), septic shock (20 (29.9%) vs. 3 (1.5%), *p* < 0.0001), and worsening kidney function (36 (53.7%) vs. 75 (36.9%), *p* = 0.015) in those who died compared to those who survived. Early surgery was performed in 54 (20%) patients overall, but less frequently in non-survivors than in survivors (3 (4.5%) vs. 51 (25.1%), *p* < 0.0001) (Table 2).

The echocardiographic data are presented in Appendix A. Transesophageal echocardiography (TOE) was performed more frequently in survivors than in deceased patients (86 (42.4%) vs. 11 (16.4%), *p* = 0.000). We found that AV-TV involvement was significantly more common in the deceased group (3 (4.5%) vs. 1 (0.5%, *p* = 0.019), and ejection fraction was significantly lower in the same group (55 (51–66) vs. 62 (55–68), *p* = 0.001).

Hemoculture results were negative in 111 (41.1%), and the most common pathogens were staphylococci in 89 (33%). Enterococci were more prevalent (25 (9.3%)) than streptococci (21 (7.8%)). GNB non-HACEK (*Hemophilus species*, *Actinobacillus*, *Cardiobacterium*, *Eikenella*, or *Kingella*) were significantly more frequent in the deceased group (10 (14.9%) vs. 9 (4.5%), *p* = 0.004), especially for Escherichia coli (5 (7.4%) vs. 4 (2.0%), *p* = 0.030) and Serratia marcescens (3 (4.5%) vs. 1 (0.5%), *p* = 0.019) (Appendix A).

Predictors of in-hospital mortality from univariate analysis are summarized in Table 3.

Logistic regression was performed on the entire sample (*n* = 270) for the period 2005–2021 to determine the simultaneous influence of significant independent variables identified in the univariate analysis to predict belonging to one of two mutually exclusive categories (alive/dead) of the dependent variable output. The logistic regression model was statistically significant, χ2(4) = 138.07, *p* < 0.0001. The model explained 59.4% (Nagelkerke R2) of the variation in outcome and correctly classified 86.3% of cases. The presence of septic shock increased the odds of death by a factor of 83.1 and the complication of acute HF by a factor of 24. Early surgery and low CCI reduce this risk (Table 4).

Logistic regression was performed on the sample (*n* = 119), covering the period 2005–2015, to determine the concurrent influence of significant independent variables identified in the univariate analysis to predict belonging to one of two mutually exclusive categories (alive/dead) of the dependent variable outcome. The logistic regression model was statistically significant, χ2(4) = 52.16, *p* < 0.0001. The model explained 52.4% (Nagelkerke R2) of the variation in outcome and correctly classified 86.6% of the cases. The probability of a fatal outcome is increased by the presence of (1) septic shock by a factor of 76.5; (2) acute heart failure complications by a factor of 11.5; and (3) to a lesser chronic heart failure extent by a comorbidity. A low CCI reduces this risk (Table 4).

Logistic regression was performed on the sample (*n* = 151) spanning the period 2016–2021 to determine the concurrent influence of significant independent variables identified using univariate analysis to predict belonging to one of two mutually exclusive categories (alive/dead) of the dependent variable output. The logistic regression model was statistically significant, χ2(5) = 118.47, *p* < 0.0001. The model explained 80.9% (Nagelkerke R2) of the variation in the outcome and correctly classified 94.0% of the cases. The most influential variable for the model (B = 5.48) was not performing early surgery in indicated patients, which increased the probability of death by 240-fold, followed by the complications of HF (B = 4.28) and septic shock (2.86) with an increased risk by 72.40 times and by 17.41, respectively. Reduced ejection fraction and previous stroke also increased the odds of death by a factor of 1.11 and 9.2, respectively (Table 4).

A survival analysis with a log-rank test was performed to determine whether there was a difference in the distribution of survival between patients indicated for early surgery that was not performed and all others. The test result is statistically significant, i.e., the survival distribution between the two groups was different—χ2(1) = 91.47, *p* < 0.0001 (Figure 1). Descriptively, the outcome is presented by the median time to event (death), with the estimated time to death being 6 days after hospitalization for the group of patients indicated for early surgery that was not performed.

A subsequent survival analysis was performed with a log-rank test to determine whether there was a difference in the distribution of survival between patients indicated for early surgery and those who did not. The result of the test was statistically significant, i.e., the survival distribution between the two groups was different—χ2(1) = 25.20, *p* < 0.0001 (Figure 2). Descriptively, the outcome is presented by the median time to event (death), with the estimated time to death being 6 days after hospitalization for the group of patients indicated for early surgery that was not performed. In addition, the 75% percentile of the data showed survival 3 days from hospitalization.

## 4. Discussion

Our sample had an overall in-hospital mortality rate for IE of 24.4%. These results are similar to those published in the literature, which ranges from 15 to 30% [2,3,21,22]. Patients in our country are generally health insured and have access to quality healthcare. The results of the univariate analysis related to increased mortality (Table 3) can be stated as follows:Patient characteristics include age, high CCI > 3, chronic renal disease, prior stroke, and atrial fibrillation.Complications include acute heart failure, renal failure, and septic shock.Echocardiographic findings include low EF (%), significant tricuspid regurgitation, and bivalve IE with aortic and tricuspid valve involvement.Microbiological characteristics of non-HACEK Gram-negative bacteria, including Escherichia coli and Serratia marcescens.Failure to undertake early surgery when required.

Gram-negative bacteria non-HACEK (Hemophilus species, Actinobacillus, Cardiobacterium, Eikenella, or Kingella) were significantly more prevalent in the deceased group, particularly Escherichia coli and Serratia marcescens. Marco Falcone et al. reported Escherichia coli to be the most common cause of GNB-IE [23]. GNB non-HASEK-IE is a rare infection that causes substantial in-hospital mortality and is defined by its presence in elderly patients with severe comorbidities, nosocomial acquisition, and a poor outcome [24,25].

Acute heart failure and septic shock were significant predictors of in-hospital death, although early surgery and a low CCI were protective. Acute cardiac failure is the most prevalent consequence of IE and the primary reason for early surgery [9,10]. The incidence of cardiac failure in IE has been reported to be between 19 and 75% [26,27]. This complication is most typically caused by leaflet perforation or rupture, mitral chordal rupture, valve dehiscence in PVIE, and, less frequently, intracardiac fistula, valve obstruction, or myocardial infarction due to embolization [10]. The intensity of presentation is determined by previous cardiac function and associated comorbidities. Heart failure in IE is a well-established independent predictor of in-hospital and one-year death. Surgery remained the only effective treatment associated with increased survival [26,27,28].

Septic shock is a life-threatening consequence of IE that affects roughly 5–12% of patients [2,3,29,30,31,32]. It is a widely recognized independent predictor of in-hospital death. *S. aureus* and Gram-negative bacteria, nosocomial acquisition, severe renal failure, diabetes mellitus, and central nervous system embolism all increase the risk of septic shock [29]. Approximately two-thirds of individuals with IE who develop septic shock die. Septic shock is the most common reason for early surgery in cases of persistent infection. There is compelling evidence that early surgical intervention lowers in-hospital and 1-year mortality in these patients [29].

Comorbidities are an essential part of the IE patient profile and a predictor of disease prognosis. We used the Charlson Comorbidity Index (CCI) [19], which is the most commonly accepted assessment of the prognostic impact of many chronic conditions. Our results are consistent with prior research on the CCI as an independent prognostic factor in IE [33,34,35].

The independent predictors of death identified in our investigation are consistent with the literature. The most generally documented predictors of death include heart failure, age > 70 years [2,3,31,36], septic shock [30,37,38], and a high CCI [2,39]. Other studies have identified PVIE, Staphylococcus aureus IE, cerebrovascular sequelae, and paravalvular abscess as independent predictors of in-hospital death [3,21,31,36,40].

Surgical intervention in IE is increasingly recommended by American and European guidelines for complicated infective endocarditis [10]. The number of patients receiving surgical treatment ranges from 20% to 70%, depending on the country and availability of cardiac surgical resources [41]. In our analysis, early surgery was performed in 20% of cases, which is consistent with the Denmark study’s 22.5% [42] and relatively low when compared to 45–62% in other countries [2,3,21,40]. Early surgery is a protective indicator, and failure to perform early surgery when indicated is a strong predictor of in-hospital death [2]. Data from the literature show that almost half of patients with IE are indicated for early surgery, but more than half of them do not receive it. The reasons are various. EURO-ENDO reports the following reasons for not performing early surgery—58.2%; death before surgery—22.5%; patient refusal—18.8%; neurological complications—11.2%; lack of cardiac surgery in the medical facility—6.2%; other—20.7% [2]. Our data demonstrate that early surgery improves survival in IE. We discovered that performing early surgery was an independent predictor of decreasing in-hospital mortality, whereas failing to conduct surgery when recommended was a substantial predictor of death in IE.

## 5. Limitation

This retrospective analysis used data from a single center’s clinical database. Another restriction is the long study period, as well as changing guidelines and evidence, particularly about early surgery, the use of novel imaging techniques for diagnosis, such as 18F-fluorodeoxyglucose positron emission tomography/computed tomography (18F-FDG PET/CT), and new antibiotic molecules. Technological developments in echocardiography have an impact on echocardiographic data. Despite these limitations, our study is the only one of its kind in Bulgaria in recent decades, and it followed a significant number of patients for a long time (17 years).

## 6. Conclusions

Acute cardiac failure and septic shock are independent predictors of in-hospital mortality. A low Charlson comorbidity index and timely surgery improved survival. Knowing the predictors of mortality would shift the therapeutic approach to be more aggressive, improving the short- and long-term prognosis of patients with IE.

## Figures and Tables

**Figure 1 microorganisms-12-01919-f001:**
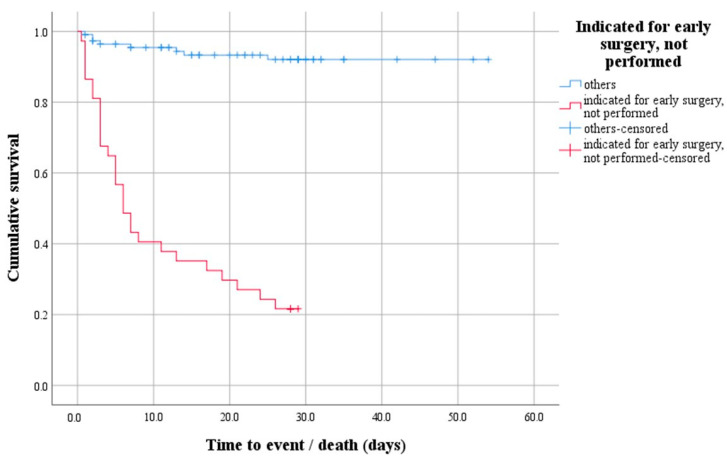
Survival function demonstrating the difference in survival distribution between patients indicated for early surgery that was not performed and all others.

**Figure 2 microorganisms-12-01919-f002:**
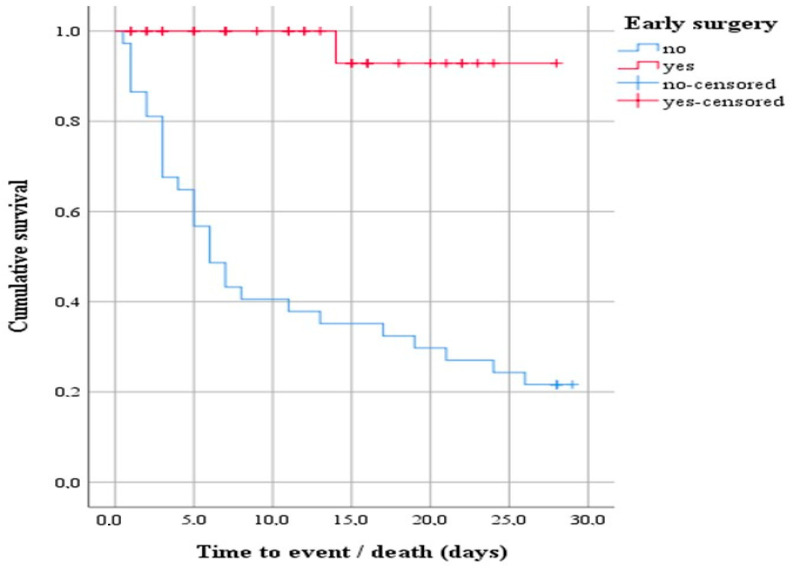
Survival function showing the difference in the distribution of survival between patients indicated for early surgery that underwent early surgery and those that did not.

**Table 1 microorganisms-12-01919-t001:** Baseline characteristics.

Variables	Total IE Cases 2005–2021 *n* = 270	IE Non-Survivors *n* = 67	IE Survivors *n* = 203	*p*-Value
Age in yrs., Χ ± SD *median (IQR)*	60.86 ± 16.83 65 (51–74)	65.99 ± 15.69 69 (59–78)	59.17 ± 16.89 63 (46–73)	0.003 †
Gender—male, *n* (%)	177 (65.6)	41 (61.2)	136 (67)	0.386 *
Time symptoms-hospitalization, *median (IQR)*	30 (20–60)	30 (14–60)	30 (20–60)	0.188 †
Risk groups, *n* (%)				
Low	136 (50.4)	36 (53.7)	100 (49.3)	0.532 *
Moderate	44 (16.3)	10 (14.9)	34 (16.7)	0.729 *
High	90 (33.3)	21 (31.3)	69 (34.0)	0.684 *
Type of valves, *n* (%)				
Native IE	180 (66.7)	45 (67.2)	135 (66.5)	0.916
Prosthetic IE	88 (33.6)	20 (29.9)	68 (33.5)	0.586
Early prosthetic	9 (3.3)	1 (1.5)	8 (3.9)	0.341
Late prosthetic	79 (29.3)	19 (28.4)	60 (29.6)	0.852
CDRIE	2 (0.7)	2 (3.0)	0 (0.0)	0.013 *
Entry door, *n* (%)				
Unknown	125 (46.3)	35 (52.2)	86 (42.4)	0.162
Non-dental manipulation/procedures	44 (16.3)	9 (13.4)	35 (17.2)	0.465
Dental	30 (11.1)	5 (7.5)	25 (12.3)	0.278
I.v. drug users	24 (8.9)	7 (10.4)	17 (8.4)	0.618
Hemodialysis	13 (4.8)	5 (7.5)	8 (3.9)	0.232
Skin	10 (3.7)	3 (4.50)	7 (3.4)	0.678
Urogenital	9 (3.3)	1 (1.5)	8 (3.9)	0.341
Gastrointestinal	5 (1.9)	1 (1.5)	8 (3.9)	0.341
Respirators	5 (1.9)	0 (0.0)	5 (2.5)	0.191
Ear Nose Throat	4 (1.5)	1 (1.5)	3 (1.5)	1.000
Others	1 (0.4)	0 (0.0)	1 (0.5)	0.562
Predisposing heart conditions, *n* (%)				
Prosthetic valve	76 (28.2)	19 (28.3)	57 (28.0)	0.949
Past IE	20 (7.4)	6 (9.0)	14 (6.9)	0.569
Past IE prosthetic	14 (5.2)	3 (4.5)	11(5.4)	0.773
Past IE native valves	6 (2.2)	3 (4.5)	3 (1.5)	0.151
Rheumatic heart disease	11 (4.0)	1 (1.5)	10 (4.9)	0.221
Congenital heart disease	21 (7.8)	6 (9.0)	15 (7.5)	0.693
Bicuspid Ao valve	11 (4.1)	4 (6.0)	7 (3.5)	0.372
Mitral valve prolapse	8 (3.0)	1 (1.5)	7 (3.5)	0.406
Others	2 (0.7)	1 (1.5)	1 (0.5)	0.410
Degenerative valve	19 (7.0)	7 (10.5)	12 (5.9)	0.202
Intact valves	123 (45.6)	28 (41.7)	95 (46.8)	0.467
Type of acquisition				
Community-acquired IE	173 (64.1)	43 (64.2)	130 (64.0)	0.976
Healthcare-associated IE	72 (26.7)	17 (25.4)	55 (27.1)	0.785
Intravenous drug use-related IE	25 (9.3)	7 (10.4)	18 (8.9)	0.714

* z-test; † Mann–Whitney U Test; CDRIE—cardiac device related IE.

**Table 2 microorganisms-12-01919-t002:** Comorbidity, clinical symptoms, and complications.

Variables	Total IE Cases 2005–2021 *n* = 270	IE Non-Survivors *n* = 67	IE Survivors *n* = 203	*p*-Value
Comorbidity				
CCI, *median (IQR)*	3 (2–5)	4 (3–6)	3 (1–5)	<0.0001 †
AH	171 (63.3)	43 (64.2)	128 (63.1)	0.860 *
CHF	124 (45.9)	32 (47.8)	92 (45.3)	0.728 *
Heart surgery	95 (35.2)	20 (29.9)	75 (36.9)	0.292 *
CKD	70 (25.9)	26 (38.8)	44 (21.7)	0.006 *
CAD	64 (23.7)	19 (28.4)	45 (22.2)	0.302 *
Diabetes	51 (18.9)	11 (16.4)	40 (19.7)	0.551 *
Atrial fibrillation	49 (18.1)	18 (26.9)	31 (15.3)	0.033 *
Past stroke	40 (14.8)	15 (22.4)	25 (12.3)	0.044 *
Gastrointestinal	32 (11.1)	6 (9.0)	26 (12.8)	0.580 *
Malignancy	30 (11.1)	6 (9.0)	24 (11.8)	0.517 *
COPD	21 (7.8)	5 (7.6)	16 (7.9)	0.936 *
Hemodialysis	14 (5.2)	5 (7.5)	9 (4.4)	0.332 *
Chronic liver disease	13 (4.8)	4 (6.0)	9 (4.4)	0.611 *
Systemic disease	4 (1.5)	0 (2.0)	4 (2.0)	0.247 *
Clinical symptoms				
Fever	263 (97.4)	63 (94)	200 (98.5)	0.045 *
Anemia	248 (92.5)	61 (91)	107 (93)	0.590 *
Cardiac murmur	178 (66.2)	46 (68.7)	132 (65.3)	0.620 *
Splenomegaly	49 (8.1)	13 (19.4)	36 (17.7)	0.759 *
Skin disorders	14 (5.5)	4 (6.0)	10 (4.9)	0.738 *
Complications				
AHF	128 (47.5)	57 (85.1)	71 (35)	<0.0001 *
Worsening kidney function	111 (41.1)	36 (53.7)	75 (36.9)	0.015 *
Embolism	56 (20.7)	16 (23.9)	40 (19.7)	0.370 *
Brain	29 (51.7)	8 (50.0)	21 (52.5)	0.866 *
Lung	5 (8.9)	0 (0.0)	5 (12.5)	-
Spleen	10 (17.9)	5 (31.25)	5 (12.5)	0.098 *
Skin	7 (12.5)	1 (6.25)	6 (15)	0.371 *
Musculoskeletal	2 (3.6)	1 (6.25)	1 (2.5)	0.495
Combine	3 (5.4)	1 (6.25)	2 (5)	0.851 *
Strock	30 (11.1)	9 (13.4)	21 (10.3)	0.486 *
Septic shock	23 (8.5)	20 (29.9)	3 (1.5)	<0.0001 *
Early surgery, *n* (%)	54 (20.0)	3 (4.5)	51 (25.1)	<0.0001 *

* z-test; † Mann–Whitney U Test; CCI—Charlson comorbidity index; AH—arterial hypertension; CHF—chronic heart failure; CKD—chronic kidney diseases; CAD—coronary arterial diseases; COPD—chronic obstructive pulmonary diseases; AHF—acute heart failure.

**Table 3 microorganisms-12-01919-t003:** Predictors of in-hospital mortality—*p*-value from univariate analysis.

Variables	Total IE Cases 2005–2021 *n* = 270	IE Non- Survivors *n* = 67	IE Survivors *n* = 203	*p*-Value
Age in yrs., Χ ± SD, *median (IQR)*	60.86 ± 16.83 65 (51–74)	65.99 ± 15.69 69 (59–78)	59.17 ± 16.89 63 (46–73)	0.003 †
Carlson Comorbidity Index, *median (IQR)*	3 (2–5)	4 (3–6)	3 (1–5)	<0.0001 †
AHF, *n* (%)	128 (47.5)	57 (85.1)	71 (35)	<0.0001 *
AKF, *n* (%)	111 (41.1)	36 (53.7)	75 (36.9)	0.015 *
Septic shock, *n* (%)	23 (8.5)	20 (29.9)	3 (1.5)	<0.0001 *
Atrial fibrillation, *n* (%)	49 (18.1)	18 (26.9)	31 (15.3)	0.033 *
CKD, *n* (%)	70 (25.9)	26 (38.8)	44 (21.7)	0.006 *
Past stroke, *n* (%)	40 (14.8)	15 (22.4)	25 (12.3)	0.044 *
AV-TV involvement, *n* (%)	4 (1.5)	3 (4.5)	1 (0.5)	0.019 *
EF %. медиана (IQR)	60 (54–68)	55 (51–66)	62 (55–68)	0.001 †
TR III ст, *n* (%)	12 (4.4)	6 (9.0)	6 (3.0)	0.040 *
*GNB non-HACEK, n (%),*	19 (7.1)	11 (15)	9 (4.5)	0.004 *
*Escherichia coli, n (%)*	9 (3.4)	5 (7.5)	4 (2.0)	0.030 *
*Serratia marcescens, n (%)*	4 (1.5)	3 (4.5)	1 (0.5)	0.019 *

* z-test; † Mann–Whitney U Test; AHF—Acute heart failure; AKF—Acute kidney failure; CKD—Chronic kidney diseases; AV-TV—aortic valve-tricuspid valve; EF—Ejection fraction; TR—Tricuspid regurgitation; *GNB non-HACEK-Gram negative bacteria non-HACEK—(Hemophilus species, Actinobacillus, Cardiobacterium, Eikenella, or Kingella).*

**Table 4 microorganisms-12-01919-t004:** Logistic regression for periods.

Variable	OR	95% CI	*p*
2005–2021			
Early surgery	0.030	0.005–0.164	<0.0001
Charlson comorbidity index	0.854	0.743–0.982	0.026
Septic shock	83.064	17.029–405.175	<0.0001
Acute heart failure	24.637	9.199–65.979	<0.0001
2005–2015			
Chronic heart failure	1.259	1.082–1.819	0.022
Septic shock	76.520	7.111–823.408	<0.000
Acute heart failure	11.528	2.870–46.294	0.001
Charlson comorbidity index	0.692	0.500–0.957	0.026
2016–2021			
Indicated for early surgery—not performed	239.882	23.171–2483.437	<0.0001
Septic shock	17.409	2.009–150.829	0.010
Acute heart failure	72.145	7.505–693.553	<0.0001
Ejection fraction (%)	1.106	1.034–1.184	0.004
Previous stroke	9.165	1.414–59.418	0.020

OR—odds ratio; CI—confidence interval.

## Data Availability

The original contributions presented in the study are included in the article/Appendix A, further inquiries can be directed to the corresponding author.

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
