# Peer review of "Infective Endocarditis—Predictors of In-Hospital Mortality, 17 Years, Single-Center Experience in Bulgaria"

_microorganisms, 2024, doi:10.3390/microorganisms12091919_

Round 1

Reviewer 1 Report

Comments and Suggestions for Authors

Thank you for the opportunity to review this excellent study. This is a well-timed paper to ascertain the risk of endocarditis with increasing percutaneous device therapies.

Mortality Rate (24%): This is consistent with the general in-hospital mortality rates reported in various studies, which typically range from 20-25%.

Charlson Comorbidity Index (CCI) >3: A higher CCI score is often associated with worse outcomes in many conditions, including IE. This aligns with the literature indicating that comorbidities significantly impact mortality rates.

Independent Predictors of Death: The study’s identification of early surgery as a protective factor and the failure to perform it as a predictor of in-hospital death is well-supported by existing research. Early surgical intervention is often recommended in specific cases to improve outcomes.

Gram-Negative Bacteria (GNB): The higher prevalence of GNB, particularly Escherichia coli and Serratia marcescens, in the deceased group is noteworthy. While Gram-positive bacteria are more commonly associated with IE, Gram-negative infections are known to be more severe and harder to treat, leading to higher mortality.

Congenital Heart Disease (CHD) and IE (7.8%): The prevalence of IE in patients with CHD is consistent with the literature, which indicates that individuals with CHD are at higher risk for developing IE.

Overall, study’s findings are in line with established knowledge, reinforcing the importance of early surgical intervention and the significant impact of Gram-negative bacterial infections on mortality in IE patients.

Comments:

1. Authors can give an overall picture of their percutaneous device therapies in their institution (N). and % of which had IE.

2. About ACHD, the guidelines they follow for IE prophylaxis.

3. Overall view of socio-economic status of the population served by the institutions. Health Insurance and accessibility to healthcare.

4. Practice of Dental care 

This above overall information will provide a broad perspective of the population at risk besides direct variables the authors have reported. I congratulate authors for an excellent study.

Author Response

Dear Reviewer,

    Thank you very much for taking the time to review this manuscript. Thank you for the constructive review, the referee’s comments have improved our manuscript. Thank you very much for the positive comments you made about our issue.

Please find the detailed responses below and the corresponding revisions/corrections highlighted changes in the re-submitted files. 

 Questions for General

Reviewer’s Evaluation

Response and Revisions

(Highlight in blue)

Does the introduction provide

sufficient background and

include all relevant references?

Is the research design

appropriate?

Are the methods adequately

described?

Are the results clearly presented?

Are the conclusions supported by

the results?

yes

yes

yes

yes

yes

No change

No change

No change

No change

No change

Comment

Respons

Location in revised MS

1.      Authors can give an overall  picture of their percutaneous device therapies in their institution (N) and % of which had IE.

1. This information is difficult to calculate right now. In the last 5-6 years, these procedures have become more prevalent with a wide spectrum of pacemakers,

ICD, CRT. Our two patients had pacemaker, single chamber VVI mode.

1. Line 137, Highlight in blue

2.      About ACHD, the guidelines they follow for IE prophylaxis.

2.  According to the current guidelines within the study (1998, 2007, 2015), they were all not eligible for antibiotic prophylaxis.

2. Line 139-142

Highlight in blue

3.      Overall view of socio-economic status of the population served by the institutions. Health Insurance and accessibility to healthcare.

3.      Patients in our country are  generally health insured and have access to quality health care.

4.      Line 238-240

Highlight in blue

   4.    Practice of Dental care 

4. The prevention is critical for reducing morbidity in IE. Maintaining proper oral hygiene and dental health, as well as following to current  recommendations for dental care and other procedures and manipulations, are critical for preventing IE.

4.Line 53-56

Highlight in blue

Reviewer 2 Report

Comments and Suggestions for Authors

The Authors described a single centre experience of IE over 17 years.  The overall methods and results seem sound.  

1.  Much of the predictors of poor outcomes for IE is known.  The authors should outline how this study adds to the literature.

2. There are no conclusions or study limitations.

3. Suggest use  P < 0.0001  instead of  P = 0.0000

4. Authors do not explain why there were 119 patients in the first 10 years vs 151 patients in the later 5 years.  What explains the major increases in cases per year in the later years?

Comments on the Quality of English Language

Only minor editing required. 

Author Response

Dear Reviewer,

      Thank you very much for taking the time to review this manuscript. Thank you for the constructive review, the referee’s comments have improved our manuscript. Please find the detailed responses below and the corresponding revisions/corrections highlighted changes in the re-submitted files. 

 Questions for General

 Reviewer’s Evaluation

Response and Revisions

Does the introduction provide sufficient background and include all relevant references?

Is the research design

appropriate?

Are the methods adequately

described?

Are the results clearly presented?

Are the conclusions supported by

the results?

Can be improved

Yes

Yes

Yes

Can be improved

Has been improved

N/A

N/A

N/A

Has been improved

Comments

Respons

Location in revised MS

(highlight in yellow)

1.  Much of the predictors of poor outcomes for IE is known.  The authors should outline how this study adds to the literature.

1. We accept your remark as valid. We have added the comment.

1. Line 78-81

2. There are no conclusions or study limitations.

2. We accept your remark as valid. We have added the conclusion and limitations.

2. Line 292-304

3. Suggest use  P < 0.0001  instead of  P = 0.0000

3. We accept your remark as valid. We have rewritten the date

3. Line 152,158,159161,

194, 202, 210.

Table 2, 3 and 4

4. Authors do not explain why there were 119 patients in the first 10 years vs 151 patients in the later 5 years.  What explains the major increases in cases per year in the later years?

We accept your remark as valid. We have rewritten the date.

Line 88-93

Reviewer 3 Report

Comments and Suggestions for Authors

This paper is a revision of the original one but I dont know if the authors responded adequately to the previous reviewer comments. In my opinion this is not a very innovative and interesting study and can be resumed just highlighting that the most important factor to avoid death in patients with IE is to perform early surgery. The English language has to be revised.

Some concerns:

line 50: old, not older

line 54 ICE-PCS??

line 98: what it means?

line 105 and following: dont list the criteria as numbers in parenthesis as can be confused with references

lines 195-196: although is explained  in table 3, please detail here what means GNB and HACEK

line 142: you say patients were indicated for early surgery but it was not performed, please explain and comment

Figures 1 and 2: please explain "censored"

line 268-269: for what?  IE?

line 285: Marco Falcone......???

The Discussion can be shortened highlighting that early surgery is the most important factor on the IE mortality which is the only important conclussion of the study

Comments on the Quality of English Language

English language has to be revised

Author Response

Dear Reviewer,

   Thank you very much for taking the time to review this manuscript. Thank you for the constructive review,

the referee’s comments have improved our manuscript. Please find the detailed responses below and the

corresponding revisions/corrections highlighted changes in the re-submitted files. 

 Questions for General

 Reviewer’s Evaluation

Response and Revisions

Does the introduction provide sufficient background and include all relevant references?

Is the research design

appropriate?

Are the methods adequately

described?

Are the results clearly presented?

Are the conclusions supported by

the results?

Yes

Yes

Can be improved

Yes

Yes

N/A

N/A

Has been improved

N/A

N/A

Comments

Respons

Location in revised MS

(highlight in pink)

1.  line 50: old, not older

1. We accept your remark as valid. We have added the correction (older than sixty years)

1. Line 58

2. line 54 ICE-PCS??

2. We accept your remark as valid.

2. This paragraph was dropped.

3. line 98: what it means?

3. We accept your remark as valid.

 4.  line 105 and following: dont list the criteria as numbers in parenthesis as can be confused with references

4.  We accept your remark as valid. We have rewritten the date.

4.  Line 104-111

5.   lines 195-196: although is explained  in table 3, please detail here what means GNB and HACEK

5.  We accept your remark as valid. We have written the date.

5. Line 179;

254-255

6.      line 142: you say patients were indicated for early surgery but it was not performed, please explain and comment

6.  Mortality is higher in individuals recommended for early surgery but it does not occur. Failure to implement is most commonly due to significant surgical risk and other reasons, which I comment on in the discussion section.

 6. Line 287-291

7.      Figures 1 and 2: please explain "censored"

7.   Our variable represents the last point at which the patients were known to be alive but weren't dead yet. That is why we have a censored survival time. We know that they survived at least up to that point (30 days), but we don't know when, after that point, they died or will die. That is the reason for censoring the data.

8. line 268-269: for what?  IE?

8.  Thank you for paying attention to me. Yes, for IE.

 8.  Line 242

9. line 285: Marco Falcone......???

  9. Thank you for paying  attention to me. Marco Falcone et al.

9. Line 256

10. The Discussion can be shortened highlighting that early surgery is the most important factor on the IE mortality which is the only important conclusion of the study 

10.  Has been improved

10. Discussion

Round 2

Reviewer 2 Report

Comments and Suggestions for Authors

No further comments

Comments on the Quality of English Language

OK